# Age-related disparities in clinical characteristics and outcomes of patients with severe fever with thrombocytopenia syndrome

Zhongwei Zhang[1◉], Xue Hu[2◉], Qunqun Jiang[1], Qian Du[1], Qianhui Chen[1], Xiaoping Chen[1], Zhiyong Ma[1], Mingqi Luo[1], Liping Deng[1*], Yong Xiong[1*]

1 Department of Infectious Diseases, Zhongnan Hospital of Wuhan University, Wuhan, China,
2 Department of Infectious Diseases, Tongji Hospital, Tongji Medical College and State Key Laboratory for Diagnosis and Treatment of Severe Zoonotic Infectious Disease, Huazhong University of Science and Technology, Wuhan, China

◉ These authors contributed equally to this work and share the first authorship.
* dengliping@whu.edu.cn (LD); yongxiong64@163.com (YX)

## Abstract

### Background and aim

The exploration of age-related clinical characteristics and prognosis of severe fever with thrombocytopenia syndrome (SFTS) has not been extensively addressed in current research. This study aimed to analyze the differences in clinical features and outcomes of SFTS patients across various age groups.

### Methods

Patients were assigned to four groups: those aged ≤ 54 years, those aged 55–64 years, those aged 65–74 years, and those aged ≥ 75 years. Then, their clinical data were compared.

### Results

A total of 253 patients diagnosed with SFTS were retrospectively included. Compared with patients aged < 65 years, patients aged ≥ 65 years had higher serum levels of laboratory parameters indicating liver, kidney, heart, and coagulation system injury, as well as a higher viral load. Moreover, the serum levels of procalcitonin, SAA, ferritin, IL-6, IL-10, and TNF-α were significantly higher, but the percentages and counts of CD3+ and CD3 + CD4 + lymphocytes were significantly lower in patients aged ≥ 65 years than in patients aged < 65 years. The cumulative survival rate of patients aged ≥ 65 years was also significantly lower than that of patients aged < 65 years. Univariate and multivariate logistic regression analyses identified that age ≥ 65 years was an independent risk factor for the prognosis of patients with SFTS.

**Data availability statement:** All relevant data are included within the paper and its supporting information files.

**Funding:** This work was supported by the Discipline Cultivation Funding, Zhongnan Hospital of Wuhan University (XKJS202025 to Y.X). The funders had no role in the study design, data collection and analysis, decision to publish, or preparation of the manuscript.

**Competing interests:** The authors have declared that no competing interests exist.

## Conclusion

Age significantly influences the clinical features and prognosis of patients, and age ≥ 65 years is associated with adverse outcomes in patients with SFTS.

### Author summary

We reported the differences in the clinical characteristics, including demographics, comorbidities, clinical symptoms, laboratory test results, complications, and treatment in severe fever with thrombocytopenia syndrome (SFTS) patients across four age groups. It was demonstrated that patients aged ≥65 years had a significantly lower cumulative survival rate compared to those aged <65 years, and age ≥ 65 years was an independent predictor of in-hospital mortality of patients with SFTS. Our study provided evidence of the impact of age on clinical characteristics and outcomes of SFTS, which suggested that tailored treatment strategies may be necessary among different age groups.

## Introduction

Severe fever with thrombocytopenia syndrome (SFTS) is an emerging zoonosis caused by Dabie bandavirus (SFTS virus, SFTSV). It was initially reported in the rural areas of China in 2009 with an estimated mortality rate up to 30% [1], and also emerged in Japan [2], Korea [3], and Vietnam [4]. The increasing incidence of SFTS has been linked to the expanding range of virus-carrying tick vectors and a growing number of human-to-human transmission via multiple routes, making it a globally emerging infectious disease [5,6]. Patients with SFTS generally present with fever and thrombocytopenia, along with various non-specific symptoms such as anorexia, headache, nausea, vomiting, and diarrhea [7,8]. Among critically ill patients with SFTS, the clinical symptoms can deteriorate in a short time, and most deaths result from multiorgan failure, including cardiac failure, acute respiratory distress syndrome, severe acute pancreatitis, renal failure, and disseminated intravascular coagulation [9–11].

Previous studies have reported that SFTS onset and death primarily occur in elderly patients, and disease prevalence increases significantly with age [12]. Some studies have demonstrated that advanced age is an independent risk factor for the prognosis of patients with SFTS [13,14]. It is also reported that age has been used to construct the predictive model for evaluating high-risk SFTS patients, which efficiently provides a basis for the timely initiation of intensified treatment to interrupt disease progression [15–17].

Although these studies have partially elucidated the relationship between age and adverse outcomes, to our knowledge, age-related differences in clinical characteristics, disease severity, treatment, and prognosis in SFTS have not been comprehensively characterized. Therefore, this study aimed to explore the clinical characteristics

and prognosis among different age groups, providing theoretical guidance for practical prevention and management strategies tailored to SFTS patients of varying ages.

## Patients and methods

### Ethics statement

The study was conducted according to the principles expressed in the Declaration of Helsinki and approved by the Ethics Committee of Zhongnan Hospital of Wuhan University. This study was approved by the Ethics Committee of Zhongnan Hospital of Wuhan University (2024178K). The written informed consent was waived due to the nature of the retrospective study. The results are reported according to the strengthening the reporting of observational studies in epidemiology (STROBE) guidelines.

### Patients

A total of 253 patients with SFTS admitted to the Department of Infectious Diseases, Zhongnan Hospital of Wuhan University between August 2016 and October 2024 were included in this retrospective cohort study. According to the age distribution of patients, they were stratified into four groups with a 10-year age interval, centered at 55, 65, and 75 years. The age ranges for each group were as follows: Group1 (G1) <55 years, $n = 36$; Group2 (G2) 55–64 years, $n = 75$; Group3 (G3) 65–74 years, $n = 105$; Group4 (G4) ≥ 75 years, $n = 37$ (S1 Fig).

### Inclusion and exclusion criteria

The inclusion criteria were as follows: (1) patients with fever (temperatures ≥ 37.3 °C); (2) patients with thrombocytopenia (platelet count < normal reference range); (3) patients with positive serum SFTSV RNA results confirmed by reverse transcriptase polymerase chain reaction.

Patients who met any of the following conditions were excluded: (1) presence of preterminal comorbidities (e.g., heart disease classified as New York Heart Association Class III–IV, severe chronic obstructive pulmonary disease, or chronic renal failure), (2) any other types of immunodeficiency, (3) history of malignant tumor, (4) incomplete clinical data or loss to follow-up. (5) concurrent infection with other viruses, such as coronavirus disease 2019 (COVID-19).

### Data collection

The medical records of patients with SFTS were reviewed, demographic details, comorbid conditions, including diabetes, hypertension, coronary heart disease,
and chronic obstructive pulmonary disease (COPD), symptoms, and laboratory test results including white blood cell (WBC) count and percentage, neutrophil count and percentage, lymphocyte count and percentage, platelet count, hemoglobin, alanine aminotransferase (ALT), aspartate aminotransferase (AST), total bilirubin (TBIL), albumin, alkaline phosphatase (ALP), gamma glutamyl transpeptidase (GGT), lactate dehydrogenase (LDH), total cholesterol (TC), triglyceride (TG), blood urea nitrogen (BUN), creatinine, cystatin-C, sodium, potassium, amylase, lipase, creatinine kinase (CK), creatinine kinase myocardial b fraction (CK-MB), troponin I, brain natriuretic peptide (BNP), prothrombin time (PT), international normalized ratio (INR), prothrombin activity (PTA), activated partial thromboplastin time (APTT), thrombin time (TT), fibrinogen, D-dimer, SFTSV viral load, occult blood test (OBT), C-reactive protein (CRP), procalcitonin, erythrocyte sedimentation rate (ESR), serum amyloid A (SAA), ferritin, interleukin-2 (IL-2), interleukin-4 (IL-4), interleukin-6 (IL-6), interleukin-10 (IL-10), tumor necrosis factor-α (TNF-α), interferon-γ (IFN-γ), CD3 + lymphocytes count and percentage, CD3 + CD4 + lymphocytes count and percentage, CD3 + CD8 + lymphocytes count and percentage, CD19 + lymphocytes count and percentage, CD16 + CD56 + cells count and percentage were collected. In our study, specimens were collected within 24 hours of patient admission.

Baseline functional status was evaluated using the Barthel Index (BI, range: 0–100), which measures dependence in basic activities of daily living [18]. Patients were categorized into four groups: total dependency (BI ≤ 20), severe dependency (BI 21–40), moderate dependency (BI 41–60), and mild or no dependency (BI 61–100).

Some severe complications, including respiratory failure, acute kidney injury, acute pancreatitis, shock, systemic inflammatory response syndrome (SIRS), and bacterial or fungal infections, disseminated intravascular coagulation (DIC), rhabdomyolysis, and haemophagocytic lymphohistiocytosis (HLH), were recorded during hospitalization. Diagnosis of SIRS met at least two of the following criteria: (1) temperature of >38 ˚C or <36˚C; (2) heart rate of >90 beats/minute; (3) respiratory rate of >20 breaths/minute; (4) white blood cell (WBC) count>12,000/mm$^3$ or <4000/mm$^3$, or differential count>10% immature polymorphonuclear neutrophil cells [19]. DIC was defined based on the International Society on Thrombosis and Haemostasis (ISTH) score, accordingly, a score upper or equal to 5 points was considered for the diagnosis of DIC [20]. Patients met the criteria for HLH according to the HLH-2004 criteria (5 of the 8) or had an HScore 169 [21].

The treatment prescriptions of hospitalized patients, including noninvasive mechanical ventilation, invasive mechanical ventilation, continuous renal replacement therapy, use of vasoactive drugs, use of corticosteroids, intravenous immunoglobulin treatment, and platelet transfusion, were reviewed and recorded in the cohort.

The primary outcome of this study was in-hospital mortality. For patients who survived the hospitalization, the follow-up time was precisely calculated starting from the day of admission to the hospital until the day they achieved clinical recovery, which was defined as the resolution of acute symptoms, normalization of vital signs, and the ability to perform basic daily activities without significant assistance. For non -surviving patients, the survival time was meticulously determined from the day of admission to the exact day of their death, as documented in the medical records.

### Statistical analysis

The quantitative data were shown as medians with interquartile ranges (IQR, P25-P75) for data with a non-normal distribution and compared by the Kruskal-Wallis method. The qualitative data were shown as numbers (percentages) and evaluated by the Chi-square test or Fisher's exact test. The cumulative survival rates, SFTSV RNA positivity rates, and platelet count normalization rates of patients across four age groups were evaluated using the Kaplan-Meier method and compared by the Log-rank test. Univariate and multivariate logistic regression analyses were performed to identify independent factors for in-hospital mortality of patients with SFTS. All data were analyzed with SPSS statistical analysis software version 26.0 (IBM Corp., Armonk, NY, USA). A two-sided $P$ value < 0.05 was considered statistically significant.

### Results

#### Demographics, comorbid conditions, baseline functional status, and clinical manifestations of patients across four age groups

The median age of included patients was 66 years (interquartile range [IQR], 59–72 years), and the age distribution of patients in the study was shown in Fig 1. The median age of the four groups (G1-G4) was 52 years (IQR, 51–53 years) for G1, 60 years (IQR, 57–63 years) for G2, 70 years (IQR, 68–72 years) for G3, and 79 years (IQR, 77–80 years) for G4, respectively. Demographic characteristics, comorbidities, baseline functional status, and clinical symptoms were compared among the four age groups of patients with SFTS. The findings revealed that patients in the G3 and G4 had a higher proportion of male, more prevalence of COPD, and a higher frequency of total dependency. Conversely, they exhibited a lower incidence of moderate dependency compared to patients in the G1 and G2. Regarding clinical manifestations, the presence of lethargy, chest distress, dyspnea, gingival bleeding, and myasthenia in the G3 and G4 were more frequent than those in the G1 and G2. No significant differences were observed in the prevalence of other comorbidities, baseline functional status categories, or remaining clinical symptoms among the four groups (Table 1).

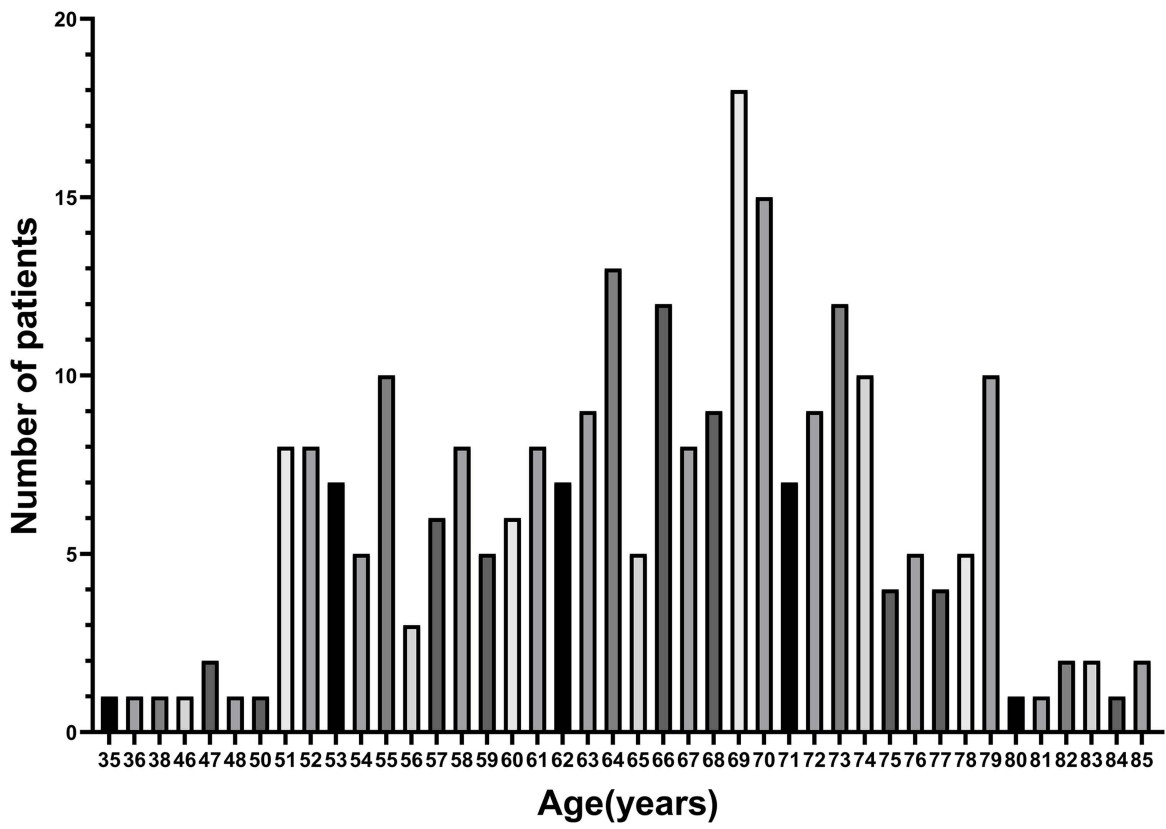

**Fig 1. The age distribution of patients in the study.**

### Laboratory test results, complications, and treatment of patients across four age groups

Among these laboratory parameters, serum levels of AST, LDH, BUN, creatinine, cystatin-C, potassium, CK, CK-MB, troponin I, and BNP of patients in the G3 and G4 were significantly higher than those of patients in the G1 and G2. In contrast, serum levels of albumin of patients in the G3 and G4 were lower than those of patients in the G1 and G2. No significant differences were observed among the four groups for the remaining laboratory variables (Table 2).

The serum levels of acute-phase proteins, inflammatory cytokines, and immune cells counts/percentages were also compared among the four age groups of patients with SFTS. As shown in Table 3, patients in the G3 and G4 had significantly higher serum levels of procalcitonin, SAA, ferritin, IL-6, IL-10, and TNF-α than patients in the G1 and G2. Additionally, absolute counts and percentages of CD3 + lymphocytes, particularly CD3 + CD4 + lymphocytes in the G3 and G4 were significantly lower than those in the G1 and G2.

In terms of complications, patients in the G3 and G4 exhibited a significantly higher prevalence of respiratory failure, acute kidney injury, shock, SIRS, bacterial or fungal infections, and DIC than those in the G1 and G2. There were no statistical differences in the incidence of acute pancreatitis, HLH, and rhabdomyolysis among the four groups. During hospitalization, patients in the G3 and G4 received a higher proportion of noninvasive mechanical ventilation, invasive mechanical ventilation, continuous renal replacement therapy, use of vasoactive drugs, and platelet transfusion compared to those in the G1 and G2. No significant differences were observed in the use of corticosteroids and intravenous immunoglobulin treatment among the four groups (Table 4).

**Table 1. Comparison of demographics, comorbid conditions, baseline functional status, and clinical symptoms of SFTS patients across four age groups.**

| | G1, <55 years (n=36) | G2, 55–64 years (n=75) | G3, 65–74 years (n=105) | G4, ≥75 years (n=37) | P value |
|---|---|---|---|---|---|
| Male, n (%) | 15(41.7) | 25(33.3) | 61(58.1) | 21(56.8) | 0.006 |
| Age (years) | 52(51-53) | 60(57-63) | 70(68-72) | 79(77-80) | <0.001 |
| **Comorbidities, n (%)** | | | | | |
| Diabetes | 3(8.3) | 13(17.3) | 9(8.6) | 4(10.8) | 0.287 |
| Hypertension | 6(16.7) | 25(33.3) | 27(25.7) | 15(40.5) | 0.098 |
| Coronary heart disease | 1(2.8) | 3(4.0) | 12(11.4) | 6(16.2) | 0.063 |
| COPD | 0 | 2(2.7) | 8(7.6) | 5(13.5) | 0.045 |
| **Baseline functional status, n (%)** | | | | | |
| Total dependency | 2(5.6) | 7(9.3) | 24(22.9) | 8(21.6) | 0.020 |
| Severe dependency | 1(2.8) | 3(4.0) | 10(9.5) | 4(10.8) | 0.282 |
| Moderate dependency | 8(22.2) | 18(24.0) | 6(5.7) | 3(8.1) | 0.017 |
| Mild or no dependency | 25(69.4) | 47(62.7) | 65(61.9) | 22(59.5) | 0.827 |
| **Clinical manifestations, n (%)** | | | | | |
| **Neurological** | | | | | |
| Coma | 0 | 1(1.3) | 7(6.7) | 3(8.1) | 0.117 |
| Lethargy | 2(5.6) | 5(6.7) | 12(11.4) | 9(24.3) | 0.027 |
| Confusion | 1(2.8) | 2(2.7) | 6(5.7) | 2(5.4) | 0.731 |
| Dysphoria | 2(5.6) | 3(4.0) | 6(5.7) | 2(5.4) | 0.962 |
| **Respiratory** | | | | | |
| Cough | 7(19.4) | 17(22.7) | 21(20.0) | 9(24.3) | 0.927 |
| Sputum | 6(16.7) | 11(14.7) | 20(19.0) | 7(18.9) | 0.882 |
| Chest distress | 6(16.7) | 22(29.3) | 48(45.7) | 16(43.2) | 0.006 |
| Dyspnea | 2(5.6) | 9(12.0) | 38(36.2) | 12(32.4) | <0.001 |
| **Gastrointestinal** | | | | | |
| Anorexia | 24(66.7) | 58(77.3) | 76(72.4) | 32(86.5) | 0.208 |
| Nausea | 18(50.0) | 47(62.7) | 68(64.8) | 21(56.8) | 0.421 |
| Vomiting | 9(25.0) | 27(36.0) | 31(29.5) | 11(29.7) | 0.655 |
| Abdominal pain | 4(11.1) | 15(20.0) | 17(16.2) | 10(27.0) | 0.308 |
| Diarrhea | 3(8.3) | 12(16.0) | 22(21.0) | 9(24.3) | 0.254 |
| **Haemorrhagic** | | | | | |
| Petechia and ecchymosis | 1(2.8) | 2(2.7) | 10(9.5) | 3(8.1) | 0.215 |
| Haematemesis | 0 | 1(1.3) | 3(2.9) | 1(2.7) | 0.704 |
| Melena | 1(2.8) | 2(2.7) | 5(4.8) | 2(5.4) | 0.839 |
| Gingival bleeding | 1(2.8) | 3(4.0) | 15(14.3) | 6(16.2) | 0.031 |
| Epistaxis | 2(5.6) | 4(5.3) | 12(11.4) | 5(13.5) | 0.335 |
| **Non-specific** | | | | | |
| Fever >38°C | 10(27.8) | 21(28.0) | 29(27.6) | 8(21.6) | 0.894 |
| Chills | 4(11.1) | 10(13.3) | 17(16.2) | 5(13.5) | 0.888 |
| Headache | 5(13.9) | 18(24.0) | 25(23.8) | 6(16.2) | 0.481 |
| Dizziness | 10(27.8) | 25(33.3) | 38(36.2) | 13(35.0) | 0.830 |
| Myalgia | 6(16.7) | 20(26.7) | 35(33.3) | 12(32.4) | 0.258 |
| Myasthenia | 5(13.9) | 16(21.3) | 44(41.9) | 15(40.5) | 0.002 |

**Table 2. Comparison of laboratory test results of SFTS patients across four age groups.**

| | G1, <55 years (n=36) | G2, 55–64 years (n=75) | G3, 65–74 years (n=105) | G4, ≥75 years (n=37) | P value |
|---|---|---|---|---|---|
| WBC ($10^9$/L) | 3.3(2.3-4.2) | 3.7(1.9-5.8) | 3.4(2.0-6.3) | 3.7(2.2-6.4) | 0.657 |
| Neutrophil (%) | 63.0(51.5-79.7) | 72.1(56.3-82.3) | 67.6(56.4-79.1) | 67.7(49.3-78.7) | 0.443 |
| Neutrophil ($10^9$/L) | 1.9(1.1-2.9) | 2.0(1.0-4.5) | 1.9(1.1-4.8) | 2.1(1.0-4.6) | 0.706 |
| Lymphocyte (%) | 28.2(14.3-35.6) | 21.6(12.0-31.8) | 23.0(12.8-31.8) | 21.4(12.0-31.6) | 0.593 |
| Lymphocyte ($10^9$/L) | 0.7(0.4-1.3) | 0.7(0.4-1.1) | 0.7(0.4-1.1) | 0.7(0.5-1.0) | 0.998 |
| Hemoglobin (g/L) | 121(111-134) | 123(112-133) | 119(109-132) | 117(110-130) | 0.763 |
| Platelet ($10^9$/L) | 48(37-78) | 45(29-68) | 42(29-61) | 44(35-59) | 0.201 |
| ALT (U/L) | 56(36-138) | 84(42-160) | 82(48-151) | 66(46-124) | 0.263 |
| AST (U/L) | 110(60-232) | 176(85-419) | 205(113-486) | 238(134-531) | 0.008 |
| TBIL(μmol/L) | 11.5(9.0-16.1) | 10.2(8.1-14.7) | 10.3(8.2-13.1) | 12.1(8.1-18.0) | 0.535 |
| Albumin (g/L) | 31.6(29.6-35.4) | 31.7(28.6-34.9) | 30.1(26.6-32.9) | 29.4(26.1-32.3) | 0.004 |
| ALP (U/L) | 71(51-99) | 72(58-95) | 75(57-111) | 69(56-81) | 0.598 |
| GGT (U/L) | 34(19-161) | 37(25-90) | 39(26-98) | 44(22-92) | 0.886 |
| LDH (U/L) | 559(325-777) | 580(391-895) | 643(511-967) | 719(635-1000) | 0.004 |
| TC (mmol/L) | 3.05(2.47-3.75) | 2.85(2.42-3.27) | 2.70(2.38-3.45) | 3.16(2.51-3.55) | 0.672 |
| TG (mmol/L) | 2.17(1.34-3.10) | 1.95(1.36-2.88) | 2.18(1.53-3.08) | 2.17(1.48-3.46) | 0.548 |
| BUN (mmol/L) | 4.5(3.1-5.9) | 5.0(3.8-6.6) | 6.4(5.1-8.4) | 6.6(5.3-9.1) | <0.001 |
| Creatinine (mmol/L) | 66(60-81) | 70(56-83) | 80(67-128) | 89(76-150) | <0.001 |
| Cystatin-C (mg/L) | 0.9(0.8-1.2) | 1.1(0.9-1.3) | 1.4(1.1-1.7) | 1.5(1.2-1.9) | <0.001 |
| Potassium (mmol/L) | 3.4(3.2-3.8) | 3.6(3.2-4.1) | 3.7(3.4-4.2) | 3.8(3.6-4.5) | 0.017 |
| Sodium (mmol/L) | 136(133-139) | 136(134-139) | 134(132-138) | 134(130-138) | 0.070 |
| Amylase (U/L) | 122(76-200) | 137(89-204) | 162(103-242) | 158(97-266) | 0.094 |
| Lipase (U/L) | 114(57-268) | 172(84-334) | 198(95-424) | 151(72-428) | 0.090 |
| CK (U/L) | 149(80-437) | 332(122-1203) | 479(181-1329) | 516(215-1428) | 0.017 |
| CK-MB (U/L) | 19(13-35) | 38(21-57) | 41(28-89) | 47(37-104) | <0.001 |
| Myoglobin(ng/ml) | 112.6(40.8-198.6) | 143.4(57.1-323.2) | 223.2(89.5-444.6) | 283.6(114.2-480.6) | 0.009 |
| Troponin I (pg/mL) | 41.8(20.3-119.3) | 87.6(36.5-250.1) | 111.3(44.9-252.8) | 125.9(60.9-287.2) | 0.002 |
| BNP (pg/mL) | 24.0(12.7-89.8) | 44.6(22.0-114.4) | 97.1(39.9-233.1) | 187.6(71.5-293.7) | <0.001 |
| PT (s) | 11.8(10.8-12.7) | 11.4(10.7-12.3) | 11.3(10.7-12.3) | 11.8(10.7-12.7) | 0.504 |
| INR | 1.08(0.99-1.17) | 1.01(0.97-1.13) | 1.04(0.98-1.13) | 1.09(1.01-1.14) | 0.578 |
| PTA (%) | 93(77-103) | 95(87-107) | 97(88-108) | 90(75-109) | 0.315 |
| APTT(s) | 38.6(34.8-44.2) | 39.5(34.0-44.9) | 41.2(35.4-49.7) | 41.0(35.3-51.8) | 0.508 |
| TT(s) | 17.5(15.8-20.3) | 18.4(16.5-21.3) | 19.4(17.2-22.9) | 19.8(17.8-23.4) | 0.018 |
| Fibrinogen(mg/dL) | 233(205-310) | 249(200-281) | 236(195-282) | 209(168-275) | 0.255 |
| D-dimer (ng/mL) | 1248(367-2456) | 954(399-3253) | 1321(584-2875) | 1170(517-2643) | 0.637 |
| Viral load ($\log_{10}$ copies/ml) | 3.5(3.3-4.1) | 3.8(3.3-4.5) | 4.2(3.4-4.6) | 4.0(3.3-4.8) | 0.094 |
| OBT positivity, n (%) | 6(16.7) | 14(18.7) | 25(23.8) | 9(24.3) | 0.714 |

## SFTSV RNA positivity rates, platelet count normalization rates, and cumulative survival rates of SFTS patients across four age groups

The SFTSV RNA positivity rates and platelet count normalization rates of SFTS patients across four age groups were compared after hospitalization, respectively.

The median duration of SFTSV RNA shedding was 10 days (IQR, 8–15 days), 12 days (IQR, 8–16 days), 15 days (IQR, 10–18 days), and 16 days (IQR, 12–20 days) of patients in the G1, G2, G3, and G4, respectively. The

**Table 3. Comparison of serum levels of acute-phase proteins, inflammatory cytokines, and immune cells counts and percentages of SFTS patients across four age groups.**

| Acute-phase proteins | G1, < 55 years (n = 21) | G2, 55–64 years (n = 42) | G3, 65–74 years (n = 58) | G4, ≥ 75 years (n = 20) | P value |
|---|---|---|---|---|---|
| CRP (mg/L) | 4.6(2.9-8.4) | 7.2(2.4-14.4) | 10.1(3.1-19.9) | 10.2(6.9-20.4) | 0.072 |
| Procalcitonin (ng/mL) | 0.10(0.05-0.34) | 0.14(0.05-0.49) | 0.24(0.09-0.73) | 0.29(0.14-0.88) | 0.043 |
| ESR (mm/h) | 7(5-14) | 8(5-14) | 8(5-13) | 8(4-14) | 0.946 |
| SAA (mg/L) | 26.3(17.1-65.6) | 46.0(25.5-107.8) | 56.7(25.5-134.7) | 62.4(29.7-158.3) | 0.016 |
| Ferritin (ng/mL) | 2660(1765-3655) | 3312(2205-4601) | 3571(2483-4923) | 3785(2549-5086) | 0.015 |
| **Inflammatory cytokines** | G1, < 55 years (n = 8) | G2, 55–64 years (n = 14) | G3, 65–74 years (n = 22) | G4, ≥ 75 years (n = 10) | P value |
| IL-2(pg/mL) | 8.5(4.5-13.2) | 7.4(6.2-14.9) | 12.6(8.3-22.5) | 13.2(8.7-26.3) | 0.128 |
| IL-4(pg/mL) | 7.2(5.6-12.4) | 6.9(4.5-17.2) | 8.6(7.1-18.4) | 8.7(6.8-24.3) | 0.475 |
| IL-6 (pg/mL) | 19.7(12.1-35.8) | 25.0(11.7-42.5) | 32.4(20.3-54.0) | 35.2(29.7-66.3) | 0.006 |
| IL-10 (pg/mL) | 10.4(9.6-19.7) | 11.2(9.3-21.5) | 18.2(12.7-25.9) | 23.7(16.5-34.8) | 0.024 |
| TNF-α (pg/mL) | 12.8(8.2-16.3) | 13.5(7.8-20.6) | 17.4(11.5-25.9) | 20.8(14.6-32.5) | 0.045 |
| IFN-γ (pg/ml) | 10.5(7.8-23.9) | 9.8(6.5-25.4) | 13.6(8.2-27.5) | 12.9(9.7-32.3) | 0.179 |
| **Immune cells** | G1, < 55 years (n = 7) | G2, 55–64 years (n = 12) | G3, 65–74 years (n = 16) | G4, ≥ 75 years (n = 8) | P value |
| CD3 + lymphocytes (%) | 64.2(58.9-77.3) | 67.3(59.5-79.4) | 59.2(47.6-71.2) | 55.7(42.8-69.4) | 0.042 |
| CD3 + lymphocytes (cells/ul) | 856(665-1875) | 892(794-1790) | 635(604-1452) | 624(583-1378) | 0.037 |
| CD3 + CD4 + lymphocytes (%) | 42.5(36.9-47.3) | 40.3(32.5-49.8) | 32.7(26.5-40.8) | 30.6(23.8-37.9) | 0.019 |
| CD3+CD4+lymphocytes (cells/ul) | 564(482-926) | 503(479-884) | 475(396-712) | 408(352-696) | 0.028 |
| CD3+CD8+lymphocytes (%) | 28.2(22.9-42.3) | 27.5(23.6-39.6) | 26.3(18.6-34.2) | 25.7(17.3-32.4) | 0.376 |
| CD3+CD8+lymphocytes (cells/ul) | 256(175-374) | 242(194-360) | 225(173-312) | 219(165-308) | 0.452 |
| CD19 + lymphocytes (%) | 8.5(7.9-17.3) | 8.3(7.5-16.4) | 7.8(6.9-15.2) | 7.6(6.6-14.1) | 0.114 |
| CD19 + lymphocytes (cells/ul) | 172(165-207) | 166(154-192) | 162(145-183) | 157(140-176) | 0.079 |
| CD16 + CD56 + cells (%) | 11.2(6.9-24.5) | 9.3(6.5-21.7) | 8.4(5.8-19.2) | 8.1(5.0-18.3) | 0.146 |
| CD16 + CD56 + cells (cells/ul) | 196(170-215) | 182(161-197) | 175(154-182) | 169(143-174) | 0.107 |

results showed that the SFTSV RNA positivity rates of patients in the G3 and G4 were significantly higher than those in the G1 and G2 after hospitalization (Fig 2A). The median duration of platelet count normalization was 16 days (IQR, 12–19 days), 17 days (IQR, 12–22 days), 19 days (IQR, 15–24 days), and 21 days (IQR, 17–26 days) of patients in the G1, G2, G3, and G4, respectively. The findings indicated that the platelet count normalization rates of patients in the G3 and G4 were significantly lower than those in the G1 and G2 after hospitalization (Fig 2B).

The cumulative survival rates of patients in the G1, G2, G3, and G4 were 94.4%, 89.3%, 74.3%, and 73.0%, respectively (Fig 3). Patients in the G3 and G4 exhibited significantly lower cumulative survival rates compared to those in the G1 and G2 (P = 0.006).

## Association of age and mortality risk in patients with SFTS

Demographics, comorbidities, clinical manifestations, baseline functional status, and laboratory parameters were included in univariate and multivariate logistic regression analyses to identify the independent risk factors for in-hospital mortality of SFTS patients. As shown in Table 5, age ≥ 65 years, total or severe dependency, neurological manifestations, dyspnea, abdominal pain, myalgia, hemoglobin, platelet count, AST, albumin, LDH, TC, BUN, creatinine, cystatin-C, potassium, amylase, lipase, CK-MB, myoglobin, BNP, PT, PTA, APTT, TT, fibrinogen, D-dimer, viral load, and OBT positivity were associated with mortality of patients with SFTS in the univariate analysis. The following factors were identified as independent predictors of in-hospital mortality: age ≥ 65 years (adjusted odds ratio (aOR), 4.019; 95% confidence interval (CI):

**Table 4. Comparison of complications and treatment of SFTS patients across four age groups.**

| | G1, <55 years (n=36) | G2, 55–64 years (n=75) | G3, 65–74 years (n=105) | G4, ≥75 years (n=37) | P value |
|---|---|---|---|---|---|
| **Complications, n (%)** | | | | | |
| Respiratory failure | 3(8.3) | 10(13.3) | 38(36.2) | 16(43.2) | <0.001 |
| Acute kidney injury | 5(13.9) | 12(16.0) | 36(34.3) | 12(32.4) | 0.010 |
| Acute pancreatitis | 4(11.1) | 14(18.7) | 28(26.7) | 9(24.3) | 0.228 |
| Shock | 3(8.3) | 8(10.7) | 32(30.5) | 12(32.4) | 0.001 |
| SIRS | 8(22.2) | 21(28.0) | 47(44.8) | 18(48.6) | 0.013 |
| Bacterial or fungal infections | 12(33.3) | 28(37.3) | 94(89.5) | 32(86.5) | <0.001 |
| DIC | 2(5.6) | 5(6.7) | 20(19.0) | 6(16.2) | 0.043 |
| HLH | 0 | 2(2.7) | 8(7.6) | 2(5.4) | 0.215 |
| Rhabdomyolysis | 7(19.4) | 20(26.7) | 36(34.3) | 12(32.4) | 0.343 |
| **Treatment, n (%)** | | | | | |
| Noninvasive mechanical ventilation | 1(2.8) | 4(5.3) | 15(14.3) | 7(18.9) | 0.034 |
| Invasive mechanical ventilation | 2(5.6) | 6(8.0) | 23(21.9) | 9(24.3) | 0.011 |
| Continuous renal replacement therapy | 1(2.8) | 4(5.3) | 14(13.3) | 7(18.9) | 0.041 |
| Use of vasoactive drugs | 2(5.6) | 5(6.7) | 30(28.6) | 12(32.4) | <0.001 |
| Use of corticosteroids | 5(13.9) | 12(16.0) | 26(24.8) | 10(27.0) | 0.268 |
| Intravenous immunoglobulin treatment | 3(8.3) | 9(12.0) | 18(17.1) | 9(24.3) | 0.208 |
| Platelet transfusion | 3(8.3) | 8(10.7) | 25(23.8) | 10(27.0) | 0.025 |

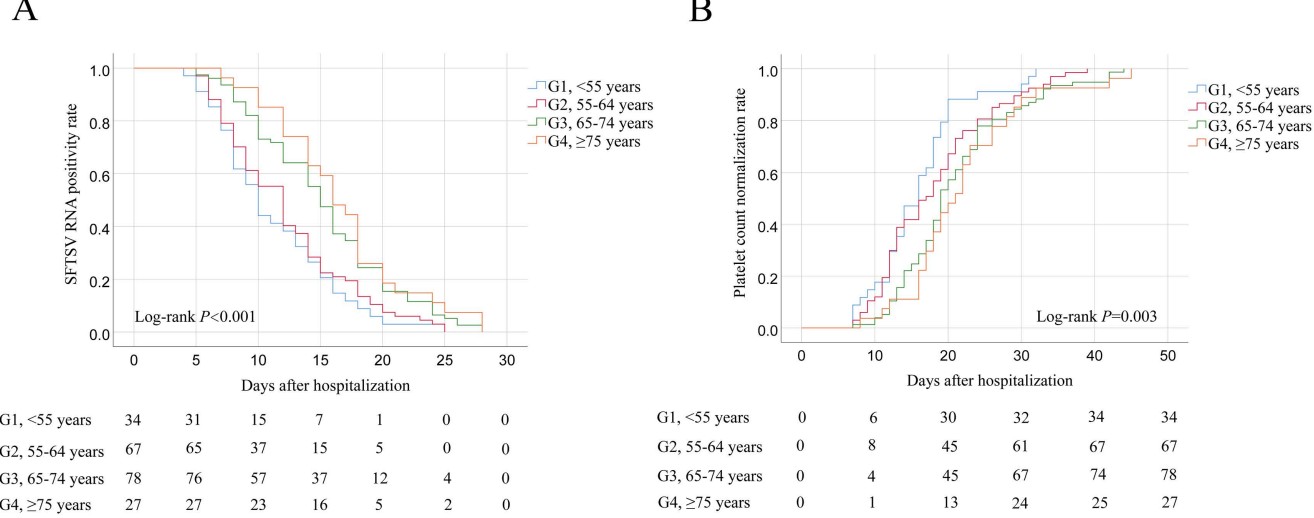

**Fig 2. A. Kaplan–Meier curves showing the SFTSV RNA positivity rates of SFTS patients across four age groups.** A comparison of the SFTSV RNA positivity rates was done using the Log-rank test, *P*<0.001. B. Kaplan–Meier curves showing the platelet count normalization rates of SFTS patients across four age groups. A comparison of the platelet count normalization rates was done using the Log-rank test, *P*=0.003.

1.568–10.304, *P*=0.004), neurological manifestations (aOR, 3.408; 95% CI: 1.415–8.211, *P*=0.006), creatinine (aOR, 1.008; 95% CI: 1.004–1.012, *P*<0.001), PT (aOR, 1.388; 95%CI: 1.024–1.882, *P*=0.034), and APTT (aOR = 1.055; 95%CI: 1.023–1.089, *P*=0.001).

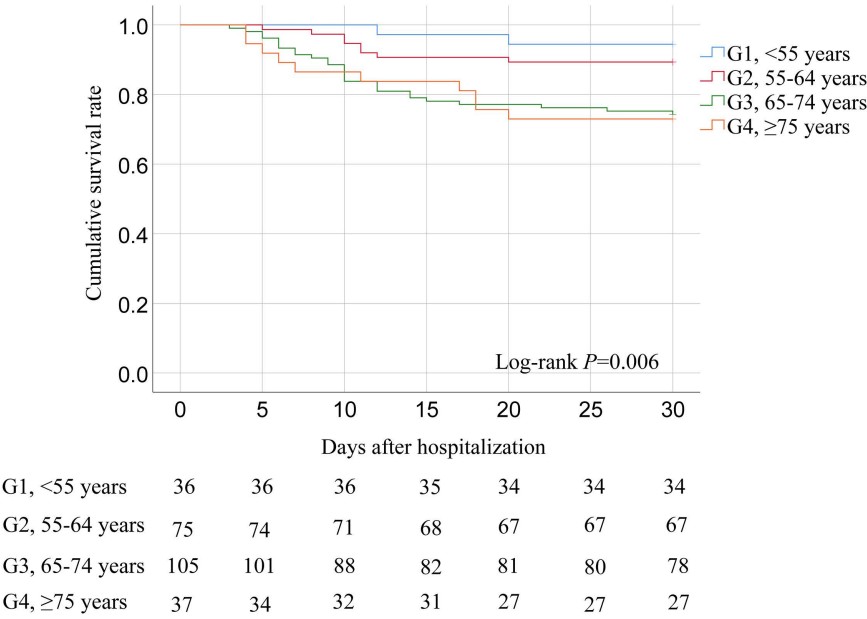

**Fig 3. Kaplan–Meier curves showing the cumulative survival rates of SFTS patients across four age groups.** A comparison of the survival estimates was performed using the Log-rank test (*P* = 0.006).

## Discussion

This study aimed to compare the clinical characteristics of SFTS patients across different age groups and to explore the impact of age on the adverse outcomes in these patients. The results demonstrated significant variations in gender, clinical manifestations, and laboratory parameters of patients with SFTS across different age groups. Most importantly, we revealed that both serious complications and poor prognosis risks also showed significant increases with age, especially in patients aged ≥65 years. It was also shown that age ≥65 years was an independent risk factor for in-hospital mortality of patients with SFTS. These findings provide a theoretical framework for considering the effects of age on the formulation of primary treatment strategies for SFTS.

In the present study, patients were assigned to four age groups, defined by nodes of 55, 65, and 75 years, based on the age distribution of the included SFTS cases. We revealed that 85.7% of inpatients were over 55 years old, which suggested that hospitalized SFTS patients tended to be older. Consistent with our data, some studies have reported that approximately 80% of SFTS occur in individuals over 50 years old, with SFTS prevalence increasing with age [22,23]. The demographic features of the four groups were different. Our work revealed male predominance in older patients (age ≥ 65 years). Therefore, extra attention is needed in the older population in clinical practice, especially for elderly men.

SFTS is a systemic infectious disease characterized by multiple-site involvement [9,24]. In our cohort, compared with younger patients (age < 65 years), older patients (age ≥ 65 years) showed more presence of lethargy, chest distress, dyspnea, gingival bleeding, and myasthenia, which generally reflect multiple organ dysfunction. A comparison of laboratory test results across four age groups also revealed that older patients had higher serum levels of laboratory parameters indicating liver, kidney, cardiac, and coagulation system injury. It is implied that older patients with SFTS are more prone to multiple organ injury or even failure.

In the present study, it was shown that older patients had higher serum levels of procalcitonin, SAA, ferritin, IL-6, IL-10, and TNF-α than younger patients. Some investigations have confirmed that procalcitonin and ferritin can be independent risk factors for the prognosis of patients with SFTS [25,26]. Chen et al. reported that serum levels of

**Table 5. Univariable and multivariable logistic regression analyses of in-hospital mortality in patients with SFTS.**

| Variables | Univariable analysis | | Multivariable analysis | |
|---|---|---|---|---|
| | OR (95% CI) | *P* value | Adjusted OR (95% CI) | *P* value |
| Male | 1.352(0.936-2.984) | 0.076 | | |
| Age ≥ 65 years | 3.559(1.681-7.535) | <0.001 | 4.019(1.568-10.304) | 0.004 |
| **Comorbidities** | | | | |
| Diabetes | 1.289(0.490-3.395) | 0.607 | | |
| Hypertension | 1.196(0.603-2.371) | 0.608 | | |
| Coronary heart disease | 1.345(0.648-2.059) | 0.712 | | |
| COPD | 1.589(0.720-4.163) | 0.594 | | |
| **Baseline functional status** | | | | |
| Total dependency | 4.218(2.015-8.720) | <0.001 | | |
| Severe dependency | 3.199(1.629-10.824) | 0.003 | | |
| Moderate dependency | 0.824(0.512-2.457) | 0.324 | | |
| Mild or no dependency | 0.645(0.312-1.058) | 0.795 | | |
| **Clinical manifestations** | | | | |
| Neurological manifestations | 5.363(2.627-10.948) | <0.001 | 3.408(1.415-8.211) | 0.006 |
| Cough | 1.761(0.862-3.596) | 0.121 | | |
| Sputum | 2.079(0.990-4.366) | 0.053 | | |
| Chest distress | 1.617(0.896-2.112) | 0.067 | | |
| Dyspnea | 2.308(1.476-3.945) | 0.032 | | |
| Anorexia | 1.521(0.711-3.253) | 0.280 | | |
| Nausea | 1.627(0.809-3.270) | 0.172 | | |
| Vomiting | 0.829(0.410-1.676) | 0.602 | | |
| Abdominal pain | 2.806(1.378-5.716) | 0.004 | | |
| Diarrhea | 1.462(0.733-2.919) | 0.281 | | |
| Haemorrhagic manifestations | 1.946(0.757-5.001) | 0.167 | | |
| Fever >38°C | 1.610(0.814-3.186) | 0.171 | | |
| Chills | 2.065(0.918-3.780) | 0.068 | | |
| Headache | 1.195(0.547-2.612) | 0.656 | | |
| Dizziness | 1.084(0.860-2.083) | 0.808 | | |
| Myalgia | 3.722(1.285-5.487) | 0.032 | | |
| Myasthenia | 1.680(0.924-2.072) | 0.063 | | |
| **Laboratory parameters** | | | | |
| WBC ($10^9$/L) | 0.971(0.875-1.078) | 0.582 | | |
| Neutrophil (%) | 1.000(0.995-1.006) | 0.897 | | |
| Neutrophil ($10^9$/L) | 0.980(0.869-1.107) | 0.749 | | |
| Lymphocyte (%) | 0.985(0.960-1.009) | 0.223 | | |
| Lymphocyte ($10^9$/L) | 0.800(0.451-1.420) | 0.446 | | |
| Hemoglobin (g/L) | 0.980(0.965-0.996) | 0.014 | | |
| Platelet ($10^9$/L) | 0.983(0.969-0.998) | 0.022 | | |
| ALT (U/L) | 1.001(1.000-1.003) | 0.130 | | |
| AST (U/L) | 1.001(1.001-1.002) | <0.001 | | |
| TBIL(μmol/L) | 1.011(0.995-1.027) | 0.187 | | |
| Albumin (g/L) | 0.855(0.790-0.926) | <0.001 | | |
| ALP (U/L) | 1.003(1.000-1.006) | 0.083 | | |
| GGT (U/L) | 1.001(1.000-1.003) | 0.145 | | |
| LDH (U/L) | 1.001(1.000-1.002) | <0.001 | | |

*(Continued)*

**Table 5.** (Continued)

| Variables | Univariable analysis | | Multivariable analysis | |
|---|---|---|---|---|
| | OR (95% CI) | *P* value | Adjusted OR (95% CI) | *P* value |
| TC (mmol/L) | 0.544(0.320-0.927) | 0.025 | | |
| TG (mmol/L) | 1.162(0.955-1.413) | 0.133 | | |
| BUN (mmol/L) | 1.134(1.069-1.202) | <0.001 | | |
| Creatinine (mmol/L) | 1.010(1.006-1.014) | <0.001 | 1.008(1.004-1.012) | <0.001 |
| Cystatin-C (mg/L) | 2.096(1.354-3.246) | 0.001 | | |
| Potassium (mmol/L) | 2.324(1.333-4.052) | 0.003 | | |
| Sodium (mmol/L) | 1.047(0.984-1.115) | 0.148 | | |
| Amylase (U/L) | 1.002(1.001-1.004) | 0.007 | | |
| Lipase (U/L) | 1.002(1.001-1.002) | 0.001 | | |
| CK (U/L) | 1.000(1.000-1.000) | 0.122 | | |
| CK-MB (U/L) | 1.007(1.003-1.012) | 0.001 | | |
| Myoglobin(ng/ml) | 1.001(1.000-1.002) | 0.014 | | |
| Troponin I (pg/mL) | 1.000(1.000-1.000) | 0.178 | | |
| BNP (pg/mL) | 1.002(1.001-1.003) | 0.001 | | |
| PT (s) | 1.585(1.261-1.991) | <0.001 | 1.388(1.024-1.882) | 0.034 |
| INR | 0.998(0.987-1.010) | 0.774 | | |
| PTA (%) | 0.970(0.952-0.989) | 0.002 | | |
| APTT(s) | 1.074(1.043-1.106) | <0.001 | 1.055(1.023-1.089) | 0.001 |
| TT(s) | 1.066(1.023-1.110) | 0.002 | | |
| Fibrinogen (mg/dL) | 0.988(0.982-0.993) | <0.001 | | |
| D-dimer (ng/mL) | 1.000(1.000-1.000) | 0.014 | | |
| Viral load ($log_{10}$ copies/ml) | 2.547(1.819-3.567) | <0.001 | | |
| OBT positivity | 2.906(1.474-5.729) | 0.002 | | |

SAA were significantly elevated in fatal patients with SFTS [27]. Our previous reports also revealed that the serum levels of SAA were substantially high in SFTS patients with serious complications such as SIRS and acute pancreatitis [8,11]. Some studies confirmed that serum levels of IL-6, IL-10, and TNF-α were obviously elevated in the mortality group, and could be independent risk factors for the prognosis of patients with SFTS [28,29]. IL-6 is a key pro-inflammatory cytokine that plays a pivotal role in the acute-phase response. Elevated levels of IL-6 in our study can be interpreted as an indication of an active immune response against the virus. However, excessive IL-6 production may lead to a cytokine storm, which is associated with severe tissue damage and multi-organ failure, as has been observed in many viral infections [30]. IL-10 is an anti-inflammatory cytokine; an increase in IL-10 could indicate an attempt by the body to control inflammation, but it may also impair the clearance of the virus [31]. TNF-α is a central mediator of inflammation and immune cell activation, and triggers an inflammatory response at the site of infection. Excessive production of TNF-α can contribute to the development of a cytokine storm, which can cause severe tissue damage, leading to multi-organ failure [32]. All these acute-phase proteins and cytokines provided an underlying mechanism to explain why older patients were more prone to a poor prognosis than younger patients in SFTS.

Immune function is usually weakened with age, and damaged immune function can accelerate the progression of viral infectious diseases [33]. Similarly, we observed that the percentage and counts of CD3+ and CD3 + CD4 + lymphocytes were significantly reduced in older patients. CD3 + lymphocytes are a broad category of T cells, which are crucial for cell-mediated immunity. CD3 + CD4 + lymphocytes, also known as helper T cells, play a central role in

coordinating the immune response by secreting cytokines that activate other immune cells. A decrease in CD3+ and CD3 + CD4 + lymphocytes may indicate an impaired immune response, making the patients more susceptible to severe viral infection and less able to clear the virus [34]. These lymphocytes play a vital role in the induction of cellular immunity in the organisms, which could stimulate the organisms to produce an immune response against viral antigens when the organisms are infected with SFTSV [35]. Some studies unveiled that T cell subsets, including CD3+ and CD3 + CD4 + lymphocytes counts, were all associated with the disease progression and severity of patients with SFTS [36,37]. In the context of our study, immunosenescence can help explain the differences in viral pathogenesis and outcomes among different age groups. Elderly patients, who are more likely to experience immunosenescence, may have a less effective immune response against the virus, leading to a higher viral load, more severe symptoms, and a greater risk of complications. This is consistent with our observation that elderly patients in our cohort had worse clinical outcomes compared to younger patients, which accords with some studies on other hemorrhagic fever viruses [38].

To our knowledge, this is the first study to report age-related differences in platelet recovery and viral shedding in patients with SFTS. The prolonged platelet recovery and viral shedding have been proven to be correlated with the high case fatality rate of patients with SFTS [39,40]. In our cohort, the durations of platelet count recovery and serum SFTSV RNA shedding in older patients were significantly longer than in younger patients. Furthermore, we found that a considerable number of serious complications, including respiratory failure, AKI, shock, SIRS, bacterial or fungal infections, and DIC were more frequent in older patients. Previous studies have reported that these complications are independent predictors of poor prognosis in patients with SFTS [7–9,41]. It is suggested that older patients are often susceptible to life-threatening complications, resulting in adverse outcomes. We demonstrated that older patients received a higher proportion of noninvasive and invasive mechanical ventilation, continuous renal replacement therapy, use of vasoactive drugs, and platelet transfusion. However, it was shown that the cumulative survival rate of older patients was significantly lower than that of younger patients. We also confirmed that age ≥ 65 years was an independent risk factor for the in-hospital mortality of patients with SFTS.

The main limitation of our study is a single-center study design. We acknowledge that this may limit the generalizability of our findings and suggest that multi-center studies are needed to confirm our results. We also recognize the subjectivity inherent in the age stratification employed in our study, where patients were categorized into four distinct groups based on age thresholds. This stratification was primarily guided by the distribution of ages among the included SFTS patients and aimed to facilitate a meaningful comparison of clinical characteristics and outcomes across different life stages. However, we recognize that such stratification is somewhat arbitrary and may not fully capture the continuous and nuanced relationship between age and disease severity. The impact of age on SFTS severity may not be linear, and other unmeasured factors could interact with age in complex ways, affecting the prognosis. Future studies could employ more sophisticated statistical methods to explore the continuous relationship between age and SFTS outcomes. Additionally, considering other relevant demographic and clinical variables in multivariate models can help disentangle the independent effects of age on disease severity.

## Conclusion

In summary, our study revealed significant disparities in gender, clinical manifestations, laboratory test results, complications, and treatment among SFTS patients across different age groups. Notably, patients aged 65 and above were more prone to multiple organ dysfunction, impaired immune function, and serious complications, resulting in a high in-hospital mortality rate. Clinicians should attach importance to these age-specific features and adopt individualized intervention strategies to improve the prognosis of patients with SFTS. Additionally, further studies can explore the differences in risk factors for poor prognosis of SFTS patients across different age groups, providing a theoretical basis for effective therapeutic strategies tailored to different age populations.

## Supporting information

**S1 Checklist. STROBE checklist.** The filled checklist is based on the STROBE Statement-Checklist of items that should be included in reports of observational studies, developed by the STROBE Initiative, https://www.strobe-statement.org/ (DOC)

**S1 Table. The normal range of laboratory parameters.** (DOCX)

**S1 Fig. The study flow chart of the enrollment of patients.** (TIF)

## Acknowledgments

The authors wish to thank the patients for participating in this study and all the staff members at our institution.

## Author contributions

**Conceptualization:** Zhongwei Zhang, Xue Hu, Liping Deng, Yong Xiong.

**Data curation:** Zhongwei Zhang, Xue Hu, Qunqun Jiang, Qian Du, Qianhui Chen, Xiaoping Chen, Zhiyong Ma, Mingqi Luo.

**Formal analysis:** Zhongwei Zhang, Xue Hu, Liping Deng, Yong Xiong.

**Funding acquisition:** Yong Xiong.

**Investigation:** Zhongwei Zhang, Xue Hu, Qunqun Jiang, Qian Du, Qianhui Chen, Xiaoping Chen, Zhiyong Ma, Mingqi Luo.

**Methodology:** Zhongwei Zhang, Xue Hu, Liping Deng, Yong Xiong.

**Project administration:** Zhongwei Zhang, Xue Hu, Liping Deng, Yong Xiong.

**Software:** Zhongwei Zhang, Xue Hu.

**Supervision:** Liping Deng, Yong Xiong.

**Visualization:** Xue Hu.

**Writing – original draft:** Zhongwei Zhang, Xue Hu.

**Writing – review & editing:** Liping Deng, Yong Xiong.

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
