## [Decision Letter · Decision Letter 0]

22 Jun 2025

Age-related disparities in clinical characteristics and outcomes of patients with severe fever with thrombocytopenia syndrome

Dear Dr. Xiong,

Thank you for submitting your manuscript to PLOS Neglected Tropical Diseases. After careful consideration, we feel that it has merit but does not fully meet PLOS Neglected Tropical Diseases's publication criteria as it currently stands. Therefore, we invite you to submit a revised version of the manuscript that addresses the points raised during the review process.

Please submit your revised manuscript within 60 days Aug 21 2025 11:59PM. If you will need more time than this to complete your revisions, please reply to this message or contact the journal office at plosntds@plos.org. Please include the following items when submitting your revised manuscript:

We look forward to receiving your revised manuscript, however, we cannot ensure acceptance of the revised manuscript if the revised version do not met the methodological standards or address the concerns raised by the reviewers.

Kind regards,

Nam-Hyuk Cho

Academic Editor

Mabel Carabali

Section Editor

Shaden Kamhawi

co-Editor-in-Chief

Paul Brindley

co-Editor-in-Chief

**Additional Editor Comments :**

This manuscript explores the impact of age on SFTS outcomes using a moderately sized cohort, but it has some methodological issues and limited novelty comparing to prior publications as commented by reviewer 1 & 2.

The authors must submit a thoroughly revised manuscript addressing the issues raised for it to be considered for publication.

**Journal Requirements:**

At this stage, the following Authors/Authors require contributions: Zhongwei Zhang, Xue Hu, Qunqun Jiang, Qian Du, Qianhui Chen, Xiaoping Chen, Zhiyong Ma, Mingqi Luo, Liping Deng, and Yong Xiong. Please ensure that the full contributions of each author are acknowledged in the "Add/Edit/Remove Authors" section of our submission form.

2) Please ensure that all Table files have corresponding citations within the manuscript. Currently, Table 5 in your submission file inventory does not have an in-text citation. Please include the in-text citation of the table.

3) Your current Financial Disclosure states, "The author(s) received no specific funding for this work.".

However, your funding information on the submission form indicates receiving a fund. Please ensure that the funders and grant numbers match between the Financial Disclosure field and the Funding Information tab in your submission form. Note that the funders must be provided in the same order in both places as well. 

Please amend your detailed Financial Disclosure statement. This is published with the article. It must therefore be completed in full sentences and contain the exact wording you wish to be published.

1) Please clarify all sources of financial support for your study. List the grants, grant numbers, and organizations that funded your study, including funding received from your institution. Please note that suppliers of material support, including research materials, should be recognized in the Acknowledgements section rather than in the Financial Disclosure

2) State the initials, alongside each funding source, of each author to receive each grant. For example: "This work was supported by the National Institutes of Health (####### to AM; ###### to CJ) and the National Science Foundation (###### to AM)."

3) State what role the funders took in the study. If the funders had no role in your study, please state: "The funders had no role in study design, data collection and analysis, decision to publish, or preparation of the manuscript."

4) If any authors received a salary from any of your funders, please state which authors and which funders..

**Comments to the Authors:**

**Please note that one of the reviews is uploaded as an attachment.**

**Reviewers' Comments:**

Reviewer's Responses to Questions

**Key Review Criteria Required for Acceptance?**

**Methods**

-Are the objectives of the study clearly articulated with a clear testable hypothesis stated?

-Is the study design appropriate to address the stated objectives?

-Is the population clearly described and appropriate for the hypothesis being tested?

-Is the sample size sufficient to ensure adequate power to address the hypothesis being tested?

-Were correct statistical analysis used to support conclusions?

-Are there concerns about ethical or regulatory requirements being met?

Reviewer #1: The methods section of the manuscript, "Age-related disparities in clinical characteristics and outcomes of patients with severe fever with thrombocytopenia syndrome," presents several methodological concerns that limit the scientific rigor and interpretability of the findings:

Lack of a Clearly Defined Hypothesis

The study lacks a clearly articulated and testable hypothesis. While it aims to explore age-related disparities in SFTS outcomes, the objectives remain broadly descriptive, and no central research question is clearly posed to guide the analysis.

Study Design and Stratification Justification

Although a retrospective observational design is acceptable for this type of investigation, the division of patients into four age groups (≤54, 55–64, 65–74, ≥75) appears arbitrary and is not supported by a rationale or clinical precedent. This undermines the methodological robustness of the stratified analysis and raises concerns about the validity of comparisons across age groups.

Study Population and Relevance

While the study seems to include a broad range of SFTS patients, the lack of justification for the age groupings makes it unclear whether the population segmentation appropriately addresses the research aims. Clearer definitions and explanations are necessary to ensure the population structure aligns with the stated objectives.

Sample Size and Statistical Power

The manuscript does not provide information regarding the sample size determination or whether a power calculation was performed. Without this, it is difficult to assess whether the study is adequately powered to detect meaningful differences, particularly when analyzing multiple subgroups.

Statistical Analysis and Confounder Adjustment

Multivariate logistic regression was employed; however, the manuscript fails to clarify whether essential confounders such as comorbidities and baseline functional status,were adequately adjusted for. This omission limits the strength of conclusions regarding age as an independent predictor of poor outcomes. Additionally, the analysis feels largely descriptive, lacking a focused approach to statistical testing aligned with a hypothesis.

Reviewer #2: I recommend analysis of TGF‐β and the titers of neutralizing antibodies.

Reviewer #3: The manuscript lacks a clear explanation of how statistical tests were chosen. For comparing continuous variables across more than two age groups, the authors report using the Kruskal-Wallis test, but there is no justification provided for not using ANOVA.

No assessment of normality of data (e.g., Shapiro-Wilk test) is described, which is necessary to determine whether ANOVA or a non-parametric test is appropriate.

Furthermore, the manuscript does not mention any post-hoc analysis to determine which groups differ when a global test (like Kruskal-Wallis) is significant. This omission limits the interpretability of group-wise differences.

Ans also, it appears that the multivariate logistic regression model includes all variables, regardless of their significance in univariate analysis.

This approach increases the risk of overfitting, multicollinearity, and may yield misleading conclusions about independent predictors.

Best practice in multivariable modeling involves including variables based on:

- Statistical significance in univariate analysis (e.g., p < 0.10 or 0.20),

- Clinical relevance,

- Absence of multicollinearity (e.g., assessed by variance inflation factors).

Reviewer #4: 1. Does the patient only need to meet one of the criteria or meet all three criteria simultaneously?

2. Please clarify the exact time point of specimen collection relative to admission in the study protocol.

**Results**

-Does the analysis presented match the analysis plan?

-Are the results clearly and completely presented?

-Are the figures (Tables, Images) of sufficient quality for clarity?

Reviewer #1: The results section of the manuscript, "Age-related disparities in clinical characteristics and outcomes of patients with severe fever with thrombocytopenia syndrome," exhibits several important weaknesses that affect the clarity and strength of the study’s conclusions:

Consistency with Analysis Plan

The results presented do not convincingly align with a predefined analysis plan. The analysis appears largely exploratory and descriptive, rather than hypothesis-driven. Without a clear hypothesis or focused research question, the rationale behind the selection and reporting of specific variables becomes unclear, raising concerns about data dredging and selective reporting.

Clarity and Completeness of Results

While the manuscript provides a large volume of clinical and laboratory data, the presentation lacks focus. Many of the variables included do not show statistically significant differences across age groups and do not contribute meaningfully to the core findings. This overabundance of data distracts from the main message and undermines the interpretability of key outcomes. Furthermore, limited interpretation of immunological findings (e.g., cytokines and immune cell subsets) weakens the translational value of these data.

Figures and Tables Quality

The figures,especially Kaplan–Meier survival curves, are reported to require improved labeling and resolution. Additionally, abbreviations in tables (e.g., DIC, SIRS, HLH) are not consistently defined at first use, which hinders readability. Overall, while the inclusion of visual data aids in understanding, the quality, labeling, and clarity of presentation must be improved to meet publication standards.

Reviewer #2: Yes

Reviewer #3: Yes to these questions.

Reviewer #4: 1. The table 5 lacks proper citations in the main text

**Conclusions**

-Are the conclusions supported by the data presented?

-Are the limitations of analysis clearly described?

-Do the authors discuss how these data can be helpful to advance our understanding of the topic under study?

-Is public health relevance addressed?

Reviewer #1: The conclusions presented in the manuscript, "Age-related disparities in clinical characteristics and outcomes of patients with severe fever with thrombocytopenia syndrome," are only partially supported by the data, and several issues limit their validity and scientific contribution.

Support for Conclusions

While the data generally confirm that older age is associated with worse clinical outcomes in SFTS patients, a finding consistent with previous studies,the conclusions drawn do not substantially extend current knowledge. Moreover, due to insufficient adjustment for important confounders (e.g., comorbidities, functional status), the claim that age is an independent predictor of poor outcomes is weakened. The exploratory and largely descriptive nature of the analysis does not allow for strong inferential conclusions.

Description of Study Limitations

The manuscript does not adequately acknowledge or discuss key limitations. Critical issues such as:

The arbitrary age group categorization,

The descriptive rather than hypothesis-driven analysis,

Potential residual confounding,

And possible redundancy with the authors’ prior publications

are either omitted or insufficiently addressed. A more transparent and detailed discussion of these limitations is necessary for scientific rigor.

Contribution to Scientific Understanding

The discussion lacks depth in interpreting the immunological findings (e.g., cytokine profiles, immune cell subsets) and does not meaningfully explore potential biological mechanisms like immunosenescence. This limits the manuscript's ability to advance understanding of age-related pathophysiology in SFTS or offer novel insights into disease mechanisms.

Public Health Relevance

The manuscript does not clearly articulate the public health implications of the findings. While SFTS is an emerging infectious disease with high mortality in older adults, the discussion misses the opportunity to emphasize how these data might inform clinical triage, treatment prioritization, or surveillance strategies for high-risk populations.

Reviewer #2: Yes

Reviewer #3: Yes.

Reviewer #4: (No Response)

**Editorial and Data Presentation Modifications?**

Reviewer #1: (No Response)

Reviewer #2: Minor revision

Reviewer #3: (No Response)

Reviewer #4: (No Response)

**Summary and General Comments**

Reviewer #1: Thank you for the opportunity to review your manuscript entitled "Age-related disparities in clinical characteristics and outcomes of patients with severe fever with thrombocytopenia syndrome." While the topic is important and clinically relevant, the manuscript has several major limitations in its design, analysis, and interpretation that preclude its acceptance in its current form.

Major Concerns:

Lack of Novelty and Contribution

The association between older age and poor outcomes in SFTS has already been established in previous studies. This manuscript largely confirms prior findings without providing novel insights or advancing the field.

Arbitrary Age Grouping Without Justification

The division into four age groups (≤54, 55–64, 65–74, ≥75) appears arbitrary. The rationale for these cutoffs is not explained, nor is there evidence that such stratification provides clinically meaningful distinctions.

Overwhelming Volume of Data Without Focus

The manuscript presents an overwhelming number of clinical and laboratory variables, many of which do not show significant differences or add value to the conclusions. This hinders readability and dilutes the main message. The analysis feels more descriptive than hypothesis-driven.

Insufficient Control for Confounders

Although multivariate logistic regression was performed, it is unclear whether important confounders (such as comorbidities and baseline functional status) were adequately adjusted for. This weakens the strength of the conclusions, particularly regarding the role of age as an independent predictor.

Limited Interpretation of Immune Findings

The reported differences in inflammatory cytokines and immune cell subsets are not adequately interpreted. The discussion lacks depth in exploring mechanisms such as immunosenescence and their relevance to viral pathogenesis and outcomes.

Redundancy With Authors’ Prior Publications

Reviewer #2: - TGF‐β levels were lower in patients with fatal disease than in patients with nonfatal disease during the initial clinical course of SFTS (Fatal outcome of severe fever with thrombocytopenia syndrome (SFTS) and severe and critical COVID-19 is associated with the hyperproduction of IL-10 and IL-6 and the low production of TGF-β. J Med Virol. 2023 Jul;95(7):e28894).

So, could authors also show TGF‐β data in this study?

- The titers of neutralizing antibodies, play an important role in protective immunity, to SFTS virus (SFTSV) in survivors, were higher than those in non-survivor patients (Neutralizing Antibodies to Severe Fever With Thrombocytopenia Syndrome Virus Among Survivors, Non-Survivors and Healthy Residents in South Korea. Front Cell Infect Microbiol. 2021 Mar 23;11:649570).

So, could authors show these data in this study?

- In line 60, SFTS

: Could authors put the full name of SFTS?

Reviewer #3: This manuscript addresses an important and timely clinical question regarding how age impacts disease severity and outcomes in patients with severe fever with thrombocytopenia syndrome (SFTS). The authors utilize a reasonably sized cohort and stratify patients into age-based subgroups to explore clinical differences and identify potential prognostic factors.

While the study is of interest and offers clinically relevant findings, there are several methodological and statistical issues that must be addressed before the manuscript is suitable for publication.

Reviewer #4: 1. SFTSV has been renamed Dabie bandavirus

2. the transmission of person to person didn't cause the increase of patients

PLOS authors have the option to publish the peer review history of their article (what does this mean? ). If published, this will include your full peer review and any attached files.

**Do you want your identity to be public for this peer review?** For information about this choice, including consent withdrawal, please see our Privacy Policy .

Reviewer #1: No

Reviewer #2: **Yes: ** LEE, KEUN HWA

Reviewer #3: No

Reviewer #4: No

**Figure resubmission:**

**Reproducibility:**



---

## [Decision Letter · Decision Letter 1]

25 Sep 2025

Response to Reviewers
Revised Manuscript with Track Changes
Manuscript

Shaden Kamhawi

co-Editor-in-Chief

Paul Brindley

co-Editor-in-Chief

**Editor Comments:**

All reviewers have recommended minor revision, as summarized below. Please carefully revise the manuscript and provide a detailed response to each of the reviewers’ comments, with particular attention to the issues raised by Reviewer #4.

**Reviewers' comments:**

**Key Review Criteria Required for Acceptance?**

**Methods**

-Are the objectives of the study clearly articulated with a clear testable hypothesis stated?

-Is the study design appropriate to address the stated objectives?

-Is the population clearly described and appropriate for the hypothesis being tested?

-Is the sample size sufficient to ensure adequate power to address the hypothesis being tested?

-Were correct statistical analysis used to support conclusions?

-Are there concerns about ethical or regulatory requirements being met?

Reviewer #2: (No Response)

Reviewer #4: (No Response)

Reviewer #5: Line 81: Is there a specific rationale or particular consideration behind the authors' use of this age-stratification approach?

Line 122-125: The diagnostic criteria for SIRS, DIC, and HLH.

Line 131-133: Detailed information regarding the follow-up, including the methodology and time frame, is requested.

Reviewer #6: The objectives of the study were clear, but lack a clear testable hypothesis.

The study designed appropriately to address the stated objectives.

The analysis of the population feels more descriptive than hypothesis-driven.

The sample size is sufficient in this study.

Statistical analysis were correct.

There are no concerns about ethical or regulatory requirements.

**Results**

-Does the analysis presented match the analysis plan?

-Are the results clearly and completely presented?

-Are the figures (Tables, Images) of sufficient quality for clarity?

Reviewer #2: (No Response)

Reviewer #4: (No Response)

Reviewer #5: Figures 2 and 3: The number of patients should be indicated at each time point. Why are the follow-up times different?

Reviewer #6: The analysis presented match the analysis plan.

The analysis of the population feels more descriptive than hypothesis-driven.

The figures (ables, Images) are of sufficient quality for clarity.

**Conclusions**

-Are the conclusions supported by the data presented?

-Are the limitations of analysis clearly described?

-Do the authors discuss how these data can be helpful to advance our understanding of the topic under study?

-Is public health relevance addressed?

Reviewer #2: (No Response)

Reviewer #4: (No Response)

Reviewer #5: The conclusion are supported by the data presented. Other limitations should be described, such as single center based analysis.

Discussion: The authors should discuss the potential mechanisms behind the differences in susceptibility to SFTSV and in antiviral immune responses across age groups, which may also apply to other similar hemorrhagic fever viruses. Additionally, it is important to note the higher proportion of males in the two older age groups, especially since male sex is associated with a poorer prognosis following SFTSV infection. The relatively small sample size may also lead to confounding effects related to gender

Reviewer #6: The strength of the conclusions were weak without providing novel insights. The design of the manuscript did not match the conclusion very well.

The limitations of analysis are learly described.

The authors did discuss how these data can be helpful to advance our understanding of the topic under study.

The public health relevance was addressed. The author compared the immune status of different age groups, and the result was quite novel.

**Editorial and Data Presentation Modifications?**

Reviewer #2: (No Response)

Reviewer #4: (No Response)

Reviewer #5: (No Response)

Reviewer #6: I agree with the opinions of the current reviewers and there are no additional comments to add.

**Summary and General Comments**

Reviewer #2: IL-8 and IFN-α levels also were higher and TGF‐β levels were lower in patients with fatal disease than in patients with nonfatal disease during the initial clinical course of SFTS (Ref. Fatal outcome of severe fever with thrombocytopenia syndrome (SFTS) and severe and critical COVID-19 is associated with the hyperproduction of IL-10 and IL-6 and the low production of TGF-β. J Med Virol. 2023 Jul;95(7):e28894).

Could authors also show data of these cytokines (IL-8, IFN-α, and TGF‐β) in this study?

Reviewer #4: The authors didn't response my comments.

Reviewer #5: The authors described the differences in the clinical characteristics, including demographics, comorbidities, clinical symptoms, laboratory test results, complications, and treatment in severe fever with thrombocytopenia syndrome (SFTS) patients across four age groups. Certain citations in the Introduction section require greater precision, and the Discussion section would benefit from a more in-depth analysis of potential mechanisms.

Line 52-55: The increasing incidence of SFTS has been linked to the expanding range of virus-carrying tick vectors and a growing number of human-to-human transmission via multiple routes. Ref: Clin Infect Dis 2021 PMID: 33068430, Int J Infect Dis 2022, PMID: 35987469, etc.

Line 60: I recommend citing studies that include larger patient cohorts or multi-center data sources, but not from the same center.

Line 61: Please identify studies that either support or refute the association between age and SFTS incidence.

Reviewer #6: The author compared the immune status of different age groups, which was kind of novel. But the analysis of the population feels more descriptive than hypothesis-driven. The strength of the conclusions were weak without providing novel insights. The design of the manuscript did not match the conclusion very well.

PLOS authors have the option to publish the peer review history of their article (what does this mean? ). If published, this will include your full peer review and any attached files.

**Do you want your identity to be public for this peer review?** For information about this choice, including consent withdrawal, please see our Privacy Policy .

Reviewer #2: No

Reviewer #4: No

Reviewer #5: No

Reviewer #6: No

**Figure resubmission:**

**Reproducibility:** To enhance the reproducibility of your results, we recommend that authors of applicable studies deposit laboratory protocols in protocols.io, where a protocol can be assigned its own identifier (DOI) such that it can be cited independently in the future. Additionally, PLOS ONE offers an option to publish peer-reviewed clinical study protocols. Read more information on sharing protocols at https://plos.org/protocols?utm_medium=editorial-email&utm_source=authorletters&utm_campaign=protocols

---

## [Decision Letter · Decision Letter 2]

21 Oct 2025

Response to Reviewers
Revised Manuscript with Track Changes
Manuscript

Shaden Kamhawi

co-Editor-in-Chief

Paul Brindley

co-Editor-in-Chief

**Reviewers' comments:**

**Key Review Criteria Required for Acceptance?**

**Methods**

-Are the objectives of the study clearly articulated with a clear testable hypothesis stated?

-Is the study design appropriate to address the stated objectives?

-Is the population clearly described and appropriate for the hypothesis being tested?

-Is the sample size sufficient to ensure adequate power to address the hypothesis being tested?

-Were correct statistical analysis used to support conclusions?

-Are there concerns about ethical or regulatory requirements being met?

Reviewer #5: I have no more comments.

Reviewer #6: Yes

**Results**

-Does the analysis presented match the analysis plan?

-Are the results clearly and completely presented?

-Are the figures (Tables, Images) of sufficient quality for clarity?

Reviewer #5: I have no more comments.

Reviewer #6: Yes

**Conclusions**

-Are the conclusions supported by the data presented?

-Are the limitations of analysis clearly described?

-Do the authors discuss how these data can be helpful to advance our understanding of the topic under study?

-Is public health relevance addressed?

Reviewer #5: I have no more comments.

Reviewer #6: Yes

**Editorial and Data Presentation Modifications?**

Reviewer #5: I have no more comments.

Reviewer #6: Yes, they made modifications based on the reviewers' comments and provided good explanations and feedback. I recommend “Accept”.

**Summary and General Comments**

Reviewer #5: The authors should acknowledge the subjectivity of age stratification and its potential impact on the analysis in the discussion section.

Reviewer #6: I think this article provides a detailed description of the immune indicators of SFTS patients, which offers certain basis for explaining the clinical symptoms of these patients. This is very good.

PLOS authors have the option to publish the peer review history of their article (what does this mean? ). If published, this will include your full peer review and any attached files.

**Do you want your identity to be public for this peer review?** For information about this choice, including consent withdrawal, please see our Privacy Policy .

Reviewer #5: No

Reviewer #6: **Yes: ** Xiaoqin Liu

**Figure resubmission:**

**Reproducibility:** To enhance the reproducibility of your results, we recommend that authors of applicable studies deposit laboratory protocols in protocols.io, where a protocol can be assigned its own identifier (DOI) such that it can be cited independently in the future. Additionally, PLOS ONE offers an option to publish peer-reviewed clinical study protocols. Read more information on sharing protocols at https://plos.org/protocols?utm_medium=editorial-email&utm_source=authorletters&utm_campaign=protocols

---

## [Editor Report · Decision Letter 3]

29 Oct 2025

Dear Xiong,

We are pleased to inform you that your manuscript 'Age-related disparities in clinical characteristics and outcomes of patients with severe fever with thrombocytopenia syndrome' has been provisionally accepted for publication in PLOS Neglected Tropical Diseases.

Best regards,

Nam-Hyuk Cho

Academic Editor

Michael Holbrook

Section Editor

Shaden Kamhawi

co-Editor-in-Chief

Paul Brindley

co-Editor-in-Chief

---

## [Editor Report · Acceptance letter]

Dear Dr. Xiong,

We are delighted to inform you that your manuscript, " 

Age-related disparities in clinical characteristics and outcomes of patients with severe fever with thrombocytopenia syndrome," has been formally accepted for publication in PLOS Neglected Tropical Diseases.

Best regards,

Shaden Kamhawi

co-Editor-in-Chief

Paul Brindley

co-Editor-in-Chief
